# Highly Stretchable Graphene Scrolls Transistors for Self-Powered Tribotronic Non-Mechanosensation Application

**DOI:** 10.3390/nano13030528

**Published:** 2023-01-28

**Authors:** Yanfang Meng

**Affiliations:** 1State Key Laboratory of Advanced Optical Communications System and Networks, School of Electronics Engineering and Computer Science, Peking University, Beijing 100871, China; yanaimengmeng@126.com; 2Beijing Institute of Nanoenergy and Nanosystems, Chinese Academy of Sciences, Beijing 100083, China; 3University of Chinese Academy of Sciences, Beijing 100049, China

**Keywords:** tribotronic, mechanosensation, ion gel, graphene scrolls

## Abstract

Owing to highly desired requirements in advanced disease diagnosis, therapy, and health monitoring, noncontact mechanosensation active matrix has drawn considerable attention. To satisfy the practical demands of high energy efficiency, in this report, combining the advantage of multiparameter monitoring, high sensitivity, and high resolution of active matrix field-effect transistor (FET) with triboelectric nanogenerators (TENG), we successfully developed the tribotronic mechanosensation active matrix based on tribotronic ion gel graphene scrolls field-effect transistors (GSFET). The tribopotential produced by TENG served as a gate voltage to modulate carrier transport along the semiconductor channel and realized self-powered ability with considerable decreased energy consumption. To achieve high spatial utilization and more pronounced responsivity of the dielectric of this transistor, ion gel was used to act as a triboelectric layer to conduct friction and contact electrification with external materials directly to produce triboelectric charges to power GFET. This tribopotential-driving device has excellent tactile sensing properties with high sensitivity (1.125 mm^−1^), rapid response time (~16 ms), and a durability operation of thousands of cycles. Furthermore, the device was transparent and flexible with the capability of spatially mapping touch stimuli and monitoring real-time temperature. Due to all these unique characteristics, this novel noncontact mechanosensation GSFET active matrix provided a new method for self-powered E-skin with promising potential for self-powered wearable devices and intelligent robots.

## 1. Introduction

The great progress of flexible electronic skins (E-skins), epidermal electronics, and mechanosensation active matrix, with the aim of miming perceptive functions of human skin, has inspired great research interests for potential applications in robotics, prostheses, and wearable healthcare monitoring devices [1]. The above devices function through converting the external stimuli into quantified physical signals and further provide feedback instructions [2,3,4,5,6,7]. Mechanosensation matrix array are employed for imitating the human somatosensory system to perceive diversifying mechanical stimuli [8,9,10,11]. Scientists and researchers dedicated their great efforts in materials design and structures optimization [12,13,14] toward a mechanosensation active matrix array to realize high sensitivity, fast response time, mechanical flexibility, and durability multimodal sensing for practical noninvasive applications requiring long-term durability, multimodal sensing, and highly integrated, simplified fabrication processes and minimized the power consumption [15,16,17,18].

As a powerful means to monitor the physiological status of the human body, electronic skins (E-skins) based on the field-effect transistor (FET) active matrix array are capable of mimicking the functions of human skin, monitoring activity and sensing and adjusting, and demonstrating their large development horizon and research value by transducing external stimuli to electronic signals. To realize demands of wearing conformally on human skin, transparent stretchable electronic devices have inspired extensive research interests. Three main available strategies to achieve stretchability for electronic device are (1) fabricating the device matrix with stretchable interconnectors [19,20,21,22,23]; (2) designing buckled- [24], spring- [25], or mesh- [26,27] shaped device configurations; and (3) developing intrinsically stretchable devices [28]. The third strategy prevails over the other two for stability and compatibility. For developing intrinsically stretchable devices, conventional intrinsically stretchable semiconductors, in general, are organic semiconductors, which may bring about detriments of a toxic, thorny procedure. As a promising alternative, a transparent stretchable electronic star material, graphene, exhibits highly desirable properties of atomic thickness, high transparency, and high conductivity [29,30,31]. However, its application in highly stretchable applications has been limited by its susceptibility to cracking at small strains [32]. For example, Seoung-Ki Lee et al. reported a stretchable, printable, and transparent transistor based on monolithically patterned graphene films but showed only 7% stretchability. Therefore, overcoming the above limitations of graphene is of great significance for new functional stretchable transparent devices. Fortunately, Bao et al. have designed highly stretchable graphene devices by intercalating graphene scrolls (the CVD graphene sheets with PMMA/front-graphene/Cu/back-graphene structure directly immersed into the Cu film etching liquid to dissolve Cu without traditional corrosion of back graphene prior to dissolving Cu). Then, the back graphene is transformed into a scroll-shaped configuration (~1 to 20 mm long, ~0.1 to 1 mm wide, and ~10 to 100 nm high) [33], with stretchability up to 120% strain (parallel to the direction of charge transport) and the ability to retain 60% of its original current output. However, in Bao et al.’s previous work, the referred graphene scrolls were limited to being employed as electrodes, as a semiconductor channel has not been reported yet. It can be anticipated that adopting graphene scrolls as the channel material of FET for E-skin not only to take advantage of their considerable evaluated stretchability but also their other special characteristic for monitoring humans’ physical parameters. We, for the first time, demonstrated that graphene scrolls possessed higher responsiveness toward temperature variations, which unprecedentedly broadened the sensing application of the graphene-based electronic devices and removed the need for external active material for temperature monitoring with a stretchable, transparent, and nontoxic center on the device. 

Given graphene scrolls as the channel of FET and active element for E-skin, self-powering has also been emphasized to address issues of mobility restriction and undesired high energy consumption. That is, harvesting mechanical energy from the human’s body to drive personalized electronics for tactile sensing functions. Fortunately, the triboelectric nanogenerator (TENG), coupling triboelectrifriction and electrostatic induction, has been demonstrated to be a powerful means to convert mechanical energy into electricity. The high voltage output of triboelectric nanogenerators (TENG) facilitates its converting mechanical energy into electrical energy with high productivity [34,35,36,37,38], as well as its combining with sensors to monitor velocity, temperature, humidity, and the other physical parameters by means of converting movement into electrical signals [39,40,41]. The TENG can also being integrated into FET Matrix as tribotronics FET to further merge the merits of FET: multiparameter monitoring, high sensitivity, and high resolution [42,43,44,45]. With respect to the tribotronics-based active matrixes, tribopotential replaces of the gate voltage source to modulate the carrier concentration along the conductive channel, allowing a direct control drain–source current by means of the external mechanical stimulus [46,47,48]. The tribotronics shows great prospects for applications as personal healthcare [49] and human–machine interaction [50]. In our previous work, we presented a self-powered noncontact mechanosensation active matrix based on an ion gel double-layer gate dielectric GFET powered by triboelectric potential form ion gel and external friction layers [51] for detecting spatial contact distances and visualizing a 2D map. 

In this manuscript, we first report a highly stretchable graphene scrolls field-effect transistor (GSFET) for self-powered tribotronic distance–temperature dual-modes mechanosensation application. In this mechanosensation active matrix, tribopotential stemming from the dielectric layer of ion gel and other friction material acted as the gate voltage. This transparent tribopotential-driven device has extremely stretchable properties (up to 120% strain (parallel to the direction of charge transport) and kept 35% of its original current output) compared to inherent graphene and layer-stacking style graphene sheets; excellent temperature sensitivity (precision as low as 1) and tactile sensing properties including high sensitivity (1.125 mm^−1^); fast response time (~16 ms); and excellent durability (over 1000 cycles test). Combining all these merits, this graphene scrolls mechanosensation active matrix achieved spatial distance and temperature dual-mapping and achieved real-time monitoring. These outstanding performances of graphene scrolls mechanosensation active arrays not only hold great potential for noncontact sensation but also will open an avenue for advanced flexible electronics both the fundamental research and practical applications. 

## 2. Experiment

### Graphene Growth on a Cu Foil by CVD

According to previous procedures [52], monolayer graphene films of high quality were prepared on a Cu foil (thickness 25 μm, 99.8%) via CVD: first, a Cu foil (25 μm) was cleaned by a mixed solution containing 98 wt% H_2_SO_4_ and 30 wt%H_2_O_2_ for 15 min. Then, Cu foil was immersed into DI water and dried under a nitrogen flow. The Cu foil was inserted into a quartz tube prior to pumping air in the chamber. Upon the pressure in the quartz tube reaching 5 × 10^−3^ Torr, the Cu foil was annealed by flowing H_2_ (10 sccm) atmosphere and heating under 1000 °C for 30 min, under a flowing H_2_ (10 sccm) atmosphere continually; 5 sccm CH_4_ was introduced to allow graphene growth. After 30 min, the CH_4_ flow shutoff, and under the same H_2_ flow conditions, the tube was cooled to room temperature. Thus, large-area (10 × 10 cm^2^) monolayer graphene was obtained on a Cu foil. 

## 3. Preparation of the Active Matrix GFET Array

### 3.1. Electrodes Fabrication

A layer of Cr/Au as the source/drain electrodes (5 nm/50 nm) was thermal evaporation, which was prior to patterning using UV-photolithography (AZ 5214 as photo-resistor, exposure time 10 s, 275 W) and etching using Au and Cr etchants, respectively. 

### 3.2. The Graphene Patterns Fabrication

The preparation of graphene patterns involved the following: Poly(methyl methacrylate) (PMMA) supporting layer was spin-coated at 4000 rpm for 5 s onto the graphene patterns on the Cu foil. Prior to being transferred onto a PDMS substrate (thickness of 500 μm), the Cu foil was chemically etched using an aqueous 0.2 M ammonium persulfate solution. Subsequently, the supporting layer was removed by dipping into the hot acetone (~60 °C) for 5 min. To fabricate trilayer graphene scrolls, the above procedure was repeated 3 times, followed by UV-photolithography (AZ 5214 as photo-resistor, exposure time 10 s, 275 W) and oxygen plasma etching (O_2_ 20 sccm, 150 W, 5 s) of the graphene film.

### 3.3. The Ion Gel Patterns Fabrication

Ion gel gate dielectrics were patterned by a UV-photolithography. The ion gel consisting of 1-ethyl-3-methylimidazolium bis(trifluoromethyl sulfonyl)imide ((EMIM) (TFSI)) ion liquid, the poly(ethylene glycol) diacrylate (PEGDA) monomer, and the 2-hydroxy-2methylpropiophenone (HOMPP) photo-initiator (weight ratio of 90:7:3) were cast onto the patterned graphene. Under UV exposure, polymerization of PEGDA was initiated by the reaction between monomers and radicals originated from HOMPP to produce the cross-linked structure. The nonexposed areas that failed to cross-link could be washed away using DI water. The graphene area in contact with the ion gel served as the active channel (L = 300 μm and W = 50 μm), whereas the other region functioned as the source/drain electrodes.

## 4. Fabrication of the Triboelectric Generator

The fabricated TENG is based on the contact/separation between a PTFE film (pulling at 5000 V 5 min, connected with an Al electrode) and a cupper film, which was connected to the ground.

## 5. Device Characterization

UV-vis measurements were carried out on an ultraviolet spectrophotometer (HOPBA). The electrical performance and triboelectronic performance of the GFETs were measured by a Keysight B1500A Semiconductor Device analyzer. A displacement motor driven by a computer-controlled stepping motor was connected with a probe station for the triboelectronic performance test. All above measurements were conducted under ambient conditions.

## 6. Results and Discussion

Figure 1a,b show the schematic illustration of the graphene scrolls FET (GSFET) array (4 × 4 pixels) with a coplanar geometry on a stretchable substrate PDMS and the corresponding circuit diagram.

We pioneeringly adopted graphene scrolls as a semiconductor stretchable channel to substitute for conventional monolayer/multilayer graphene. The fabrication process of graphene scrolls is according to Bao et al.’s previous work. It is well-established that CVD growth of graphene on both sides of the copper film forms Gr/Cu/Gr structures. In regard to transferring graphene, a thin layer of poly(methyl methacrylate) (PMMA) was spin-coated to protect the front side of the graphene; followed by that the entire film (PMMA/front graphene/Cu/back graphene) was floated on (NH_4_)_2_S_2_O_8_ solution to etch away the Cu foil. The backside graphene without the PMMA coating inescapably had cracks and defects that allowed an etchant to penetrate through [53,54]. It was observed that the released graphene domains rolled up into scrolls originated from surface tension and immediately attached to the remaining front-G/PMMA film. The as-prepared front-G/G scrolls could be transferred onto pre-UV/O_3_-treated PDMS, and PMMA was given away by using acetone and repeating the transfer to another layer subsequently. The obtained monolayer graphene scrolls had similar characteristics to the monolayer back-etched graphene (Appendix A left panel), while there was evident discrepancy between the obtained monolayer graphene scrolls and the monolayer back-etched graphene (Appendix A right panel). As shown in Appendix A left panel, Raman spectra of the monolayer back-etched graphene (black curve) and monolayer graphene scrolls (blue curve) exhibited two characteristic peaks of G band at ~1597 cm^−1^ and ~1569 cm^−1^, respectively, and 2D band at ~2694 cm^−1^ and ~2696 cm^−1^, respectively. The monolayer characteristic of graphene was verified by both the full width at half-maximum (~29 cm^−1^) of the symmetric 2D band and the 2D/G intensity ratio (~2.5). As shown in Appendix A right panel, the Raman spectra of trilayer graphene scrolls (blue curve) have similar positions of G (1581 cm^−1^) and 2D (2687 cm^−1^) peaks as the monolayer characteristic of graphene (black curve) but with a distinguishing variation of the 2D/G intensity ratio (~0.22) for the 2D/G intensity ratio of the monolayer characteristic of graphene, which is about (~2.5). It is worth noting that no other surplus peak exists in the Raman spectra of trilayer graphene scrolls, suggesting no newly generated structure besides graphene. Ion gel (composed of 1-ethyl-3-methylimidazolium bis(trifluoromethyl sulfonyl)imide ((EMIM)(TFSI)) ion liquid, poly(ethylene glycol) diacrylate (PEGDA) monomers, and 2-hydroxy-2-methylpropiophenone (HOMPP) photo-initiator in a weight ratio of 90:7:3), acting as both the gate dielectrics of GSFET and a tribo-electrification layer, was patterned by photolithography above the graphene channel. (The specific fabrication procedure is the same as our previous work, as shown in Appendix A). The graphene layer was applied as the transistor channel (in contact with the ion gel), as well as serving as the source and drain electrodes due to its semimetal characteristic. The width and length of the graphene channel were 100 µm and 1000 µm, respectively. The microscopy image of trilayer graphene scrolls is presented in Figure 1b. The GSFET array also exhibited good optical transparency, being substantiated by UV-vis spectroscopy. As shown in Figure 1c, the PDMS film exhibited a transmittance of 91.5% in the visible and near-infrared region. In the cases of the monolayer graphene film on PDMS and graphene scrolls film with ion gel on PDMS, the transmittance are almost consistent with values of 88% and 83.2%, respectively. Photographs of the flexible GSFET array inserted in Figure 1c demonstrated its flexibility and transparency.

To demonstrate advantages of graphene scrolls as the active element for mechanosensation application, we comprehensively investigated the properties of graphene scrolls. Bao et al. justified that the scrolls are rolled graphene in nature by methods of studying on the monolayer front-G/G scroll structures by high-resolution transmission electron microscopy (TEM) and electron–energy loss (EEL) spectroscopy [33]. The atomic force microscopy (AFM) images provide insight into the monolayer graphene (Appendix A left panel) and the distribution of graphene scrolls in microstructure (Appendix A right panel). As shown in Appendix A right panel, the scrolls are arbitrarily distributed on the surface, and their in-plane density increases proportionally to the number of stacked layers. The length and the width are about 1 to 10 mm and 0.1 to 0.8 mm, respectively. By contrast, the monolayer graphene exhibits plat configuration. 

Besides structure characterization, the electrical properties of graphene scrolls were also investigated by making comparison to monolayer and layer-stacking styles of graphene (during the transferring of the graphene procedure, corrosion of back graphene prior to dissolving Cu). Different styles of graphene films, trilayer graphene scrolls, trilayer graphene, bilayer graphene scrolls, and monolayer graphene were transferred onto Ecoflex and patterned into 500 μm-wide and 2000 μm-long channels by photolithography. Two-end resistances as a function of strain deformations and bending deformations were tested under ambient conditions. As shown in Figure 2a (left panel), when strain deformations were perpendicular to the direction of current flow, trilayer graphene scrolls realized the highest relative strain before breaking: up to 120%, exceeding trilayer graphene (110%) and bilayer graphene scrolls (90%) and standing out be far superior to monolayer graphene (7%). More importantly, the resistance tolerance toward strain deformation of trilayer graphene scrolls was almost up to 120%, outweighing other styles of graphene. Figure 2a right panel depicts strain-dependent two-end resistances when strain deformations were parallel to current flow, basically the same trend as the aforementioned perpendicular to current flow. Trilayer graphene scrolls realized the highest strain rate before breaking: up to 120%, exceeding trilayer graphene and bilayer graphene scrolls (90%) (trilayer graphene and bilayer graphene scrolls showed exactly the same values) and prevailing much more over monolayer graphene (7%). The results of Figure 2a demonstrate the outstanding resistance tolerance toward strain of trilayer graphene scrolls.

To explore the underlying reason behind excellent electrostretchability properties of trilayer graphene scrolls, Raman spectra of trilayer graphene scrolls and monolayer graphene were compared under strain (Appendix A). Appendix A depicts the Raman spectra of monolayer graphene (left panel) and trilayer graphene scrolls (right panel) in free-standing state (blue curve) and strain state (green curve). It can be observed that the Raman spectra of monolayer graphene experienced more pronounced variation compared with that of trilayer graphene scrolls.

Report-referred graphene scrolls proposed by Bao et al. are limited to application as the electrode; application as the semiconductor channel has not been reported yet. We, for the first time, investigated properties of GSFET based on trilayer graphene scrolls as a semiconductor channel, demonstrating their superiority. The width and length of graphene channel of the single GFET were 50 µm and 300 µm, respectively, with a 100 µm distance between gate and channel. The *I–V* character and transfer characteristic of trilayer graphene scrolls under variation of relative strain deformation and bending deformation were compared against those of monolayer graphene. As depicted in Figure 2b (output curve of GFET), as strain was given at the magnitude of 5%, the resistance of GSFET (FET based on trilayer graphene scrolls) increased 40% while the resistance of GFET (FET based on monolayer graphene) increased 10 folds. Trilayer graphene scrolls possess good resistance retention until experiencing strain up to 31%, while monolayer graphene undergoes breakdown when strain increased to only 7%. Figure 2c shows the transfer characteristic of GSFET (top panel) and GFET (bottom panel) under varied relative strain deformation. It can be clearly observed in Figure 2c top panel that the transfer characteristic of trilayer graphene scrolls changed a little. 

Figure 2d,e display the *I–V* character and transfer characteristic of GSFET and GFET under varied relative bending deformation. The case is the same as the strain deformation: trilayer graphene scrolls possess good resistance retention toward deformation. It is worth noting that it was beyond our expectation that the transfer curves turned out to be same as when the monolayer graphene was the channel, with Dirac point being around 0.6.

Apart from comparing trilayer graphene scrolls with monolayer graphene, comparison of trilayer graphene scrolls and trilayer stacking graphene (during transferring graphene procedure, corrosion of back graphene prior to dissolving Cu) was also conducted.

As shown in Appendix A, as the GSFET undergoes 31% strain, both output characteristic and transfer characteristic of trilayer graphene scrolls displayed little changed (retaining 50% electrical properties) while trilayer graphene experienced electric breakdown as strain exceed 31%. The excellent electrostretchability properties of trilayer graphene scrolls demonstrate its potential application for E-skin. 

In practical application of E-skin, multiple functional integrations are also highly required to simultaneously perceive the realization of biomimetic function of E-skins. Given widespread of temperature responsiveness and large in-plane thermal conductivity of graphene [55], we attempted to probe into the temperature responsiveness of trilayer graphene scrolls. The *I–V* character of trilayer graphene scrolls under varied temperature were compared with that of monolayer graphene (graphene films transferred on Ecoflex were patterned into 500 μm-wide and 2000 μm-long channels by photolithography). Obviously, the trilayer graphene scrolls displayed more distinct temperature-dependence than that of the counterpart monolayer graphene: every 1 °C induced a 0.7% current variation amount (Figure 3a top panel), while monolayer graphene almost was immune to temperature variation (every 1 °C induced 0.13% current variation) (Figure 3a bottom panel). Correspondingly, we calculated the relative resistance variation (*R − R*_0_)/*R*_0_ toward temperature variation of trilayer graphene scrolls (Figure 3b top panel) and monolayer graphene (Figure 3b bottom panel), verifying the more pronounced temperature-dependence of graphene scrolls. To investigate temperature responsiveness of FET based on trilayer graphene scrolls, we used two graphene field-effect transistor: the width and length of trilayer graphene scrolls and monolayer graphene channels were 50 µm and 300 µm, respectively, with a 100 µm distance between gate and channel. 

Considering E-skin application, we conducted measurement near the human body’s core temperature. Comparison of the transfer characteristics of GSFET and GFET under variation of temperature evidently shows that the GSFET exhibits more sensitivity to temperature variation (Figure 3c). To be specific, as the *I*_DS_ is fixed at 0.5 V, *I*_DS_ of GSFET increases from 252 μA to 277 μA when the temperature increases from 25 °C to 45 °C with the relative variation (Figure 3c top panel) more distinct than that of the counterpart of GFET (*I*_DS_ of GSFET increases from 78.5 μA to 82.4 μA as the temperature increases from 25 °C to 45 °C (Figure 3b bottom panel))P. To verify feasibility of GSFET for E-skin, real-time temperature monitoring is shown in Figure 3d. Compared to the real-time temperature monitoring of GFET (right panel), GSFET (left panel) exhibited more pronounced output variation, suggesting its better feasibility for temperature monitoring. Importantly, the stability of output current upon temperature, achieving balance and reversibility, suggested GSFET as an ideal candidate for E-skin for monitoring physiological temperature. The hysteresis is one of graphene’s characteristics that is relative to defect and doping, here indicated by strain and bending deformation: the defect could occur and change graphene properties. Therefore, the hysteresis probably influences GSFET and brings about temperature fluctuation to some extent. 

To elucidate the mechanism of more pronounced temperature-dependence of GSFET than that of the counterpart GFET, Raman spectra of trilayer graphene scrolls and monolayer graphene were compared under varied temperature (Appendix A). It can be observed that Raman spectra of trilayer graphene scrolls (Appendix A right panel) underwent greater variation toward high temperature compared to that of monolayer graphene (Appendix A left panel). 

Considering the energy consumption and based on our previous work, self-powered GFET array extends to the self-powered GSFET array. It is worth noting that ion gel dielectric is directly served as the electrifriction layer to carry out friction with external materials to simplified fabrication procedure with high spatial compaction. Prior to studying GFET array, single tribotronic GSFET was comprehensively explored. The basic layout of the single FET is schematically illustrated in Figure 4a. The width and length of the graphene channel were 50 µm and 300 µm, respectively, with a 100 µm distance between gate and channel. 

Figure 4d left panel and right panel display output characteristic and transfer characteristic of tribopotential-gated GSFET by electrification between ion gel and skin, separately. Due to skin being a positively triboelectric material relative to the ion gel, *I*_D_ experienced increase gradually as the distance between ion gel and skin increased, corresponding to the GFET performing under a negative gate voltage. This linear relationship between *I*_D_ variation and contact distance can be divided into regions I and II. *I*_D_ variation behavior was divided into two parts: when the contact distance increased from 0 to 1 mm, *I*_D_ was increased from 20.0 μA to 22.5 μA with variation gradient of 2.5 μA/mm; when the contact distance increased from 1 to 3 mm, *I*_D_ was increased from 22.5 μA to 23.8 μA with gradient of 0.65 μA/mm. The 1 mm distance variation induced *I*_D_ variation of 2.5 µA, equivalent to a gate voltage of 0.66 V. The variation ratio is about 1.125 mm^−1^ (2.5 μA/mm/20.0 μA) (as the contact distance increased from 0 to 1 mm), suggesting the excellent sensitivity of the triboelectric properties and offering sound fundamentals for the sensor application.

On basis of the triboelectric series [54], electrons transferred from the skin to ion gel, leaving net negative electrostatic charges on the surface of the ion gel layer and positive electrostatic charges on the skin. The produced triboelectric charges in opposite polarities were fully balanced and had no effect on the graphene channel. As the skin leaves the surface of the ion gel, the un-offset negative charges on the surface of the ion gel attract anions in the ion gel and produce an EDL at the interface between ion gel and the skin, leaving anions at ion gel/graphene interface, corresponding to applying a negative voltage on the GFET. Consequently, the Fermi level of graphene channel was upped downward, leading to an increase in the drain current. To investigate the dynamic triboelectric, real-time tests for the triboelectronic GSFETs were performed under repeat electrification between ion gel and skin. To demonstrate the endurability for practical application, a cyclic durability test was conducted by ion gel and skin’s repeated contact–separation with triboelectric distance of 1 mm (Figure 4e left panel). One contact and separation was regarded to be one cycle. Over 3000 cycles, a consistent output current was kept during repeated triboelectrification. It is worth mentioning that the current level was kept at a constant value for a long time after stopping triboelectric termination (Figure 4e right panel). The above results support the durability and reliability of the GFET for noncontact mechanosensation E-skin practical applications.

This GSFET was compared with our previous work triboelectric GFET based on monolayer graphene (same configuration and size, friction between ion gel and PTFE as gating) (Appendix A). For GFET’s triboelectricity, the 200 μm distance variation between PTFE and ion gel induced *I*_D_ variation of 3.63 µA, equivalent to a positive gate voltage of 0.31 V, while for trilayer graphene scrolls FET’s triboelectricity, the 200 μm distance variation of between PTFE and ion gel induced *I*_D_ variation of 1.1 µA, equivalent to a positive gate voltage of 0.24 V. It was indicated that trilayer graphene scrolls as the channel was less susceptible to triboelectric.

EDLT analysis can provide deeper insight into the lower susceptibility of trilayer graphene scrolls toward triboelectric than monolayer graphene. According to ion gel EDLT’s equivalent circuit [56,57], the total capacitance *C* is defined as (1/*C*_G_ + 1/*C*_Q_)^−1^. *C*_Q_ and *C*_G_ in multilayer graphene act as a function of the layer number [58]. The value of *C*_Q_ increases with increasing layer number because of increasing density of states and saturation as the layer number reaches six. The bilayer and trilayer show anomalies in their transfer curves, which can be ascribed to a second sub-band in presence at approximately 0.5 eV above the Dirac point and is generated by interlayer interactions (the features of multilayer graphene). Appendix A shows schematic illustration of degradation of triboelectric responsiveness of trilayer graphene scroll FET. 

Compared to monolayer graphene FET, the higher-lying sub-bands are generally unreachable by conventional solid-gated FETs but were reached for the first time by utilizing ion gel (electron double layer effect EDLT), owing to the high-density charge accumulation. Taking the higher-lying sub-band into consideration, a significant capacitance increase with increasing *V*_G_ is understood.

Because the capacitance of diffuse layer is negligible, the interfacial capacitance of ion gel can be recognized as a serial combination of an electric double layer capacitance (*C*_EDL_) and a quantum capacitance of the graphene (*C*_Q_) [59]. With respect to EDLT in ion gel, the potential difference between EDL capacitance and quantum capacitance of the graphene is conformed to the following equation:(1)$$\left|V_{G}-V_{G,\min}\right|=\frac{\hbar n_{F}\sqrt{\pi n}}{e}+\frac{ne}{C}$$
(2)$$V_{G}=V-V(R)=-\frac{Q}{S\varepsilon_{0}}(d_{0}+x(t))+\frac{\sigma x(t)}{\varepsilon_{0}}-V(R)$$
where *ћ* represents the reduced Planck’s constant, *ν*_F_ is Fermi velocity (1.1106 m/s), *e* is the electron charge, and *n* is the charge density. *V*_Gmin_ is the gate voltage corresponding to Dirac point. According to Equation (2), the carrier concentration is closely associated with *V*_G_ and *C*_EDL_. *V*_G_ is the effective voltage that triboelectric is posed on; according to Equation (2), the value of *V*_G_ is equal to that of the open voltage of the TENG substrate. The resistance consumption V(R) and open voltage of TENG followed a linear function as distance between two triboelectric materials increased.

In light of Equation (1), the increment of layer number change *V*_G_ further changes the relationship of *V*_G_ and triboelectric distance, unambiguously explaining the lower susceptibility of trilayer graphene scrolls toward triboelectric than monolayer graphene.

The working principle of tribopotential characteristic of this GSFET is the same as our previous work: noncontact mechanosensation active matrix based on tribotronic planar graphene transistors array [60]. There is no need to repeat it here.

Our highly stretchable GSFET also possesses excellent rapid dynamic response. As shown in Appendix A, the *I*_D_ response time upon application and release of the triboelectric was only 15 ms, calculated from the vertical rising and dropping of *I*_D_. The rapid response time demonstrates the outstanding sensitivity of the device. The *I*_D_ increased upon the contact–separation cycles commenced between skin and the ion gel (under applied drain voltage of 0.1 V).

Given that the ultimate goal of triboelectric-driving GFET was applied for noncontact mechanosensation, GSFET array on substrate of PDMS was fabricated with a 4 × 4 GSFETs matrix configuration, as shown in the circuit diagram in Figure 5a. Prior to mapping the spatial temperature and distance applied to the active matrix, the electrical performances of the 16 GSFETs were characterized. The output sensing current signals from different sensors under presenting temperature were collected and plotted to obtain the corresponding 2D color maps. 

The triboelectric distances applied to the active matrix were spatially mapped, as shown in Figure 5b. Two edges or two corners of the matrix were fixed, and only a 0.1 V potential was applied to the bit lines. The 2D color mapping (Figure 5c) of the temperature distribution and distances distribution on the active matrix showed the potential application of monitoring spatial interaction of human skin. Figure 5d displays the real-time temperature monitoring and distance monitoring between the ion gel and skin. The stable curves demonstrate the feasibility of the device for non-mechanosensation active matrix application. As a small application, the GSFET active matrix was conformally attached onto the knee joint of the test subject. The 2D color mapping of the distances distribution (Figure 5e) and the temperature distribution (Figure 5f) on the active matrix can reflect in real-time the situation of the human knee joint. Notably, the monitored temperature is a little lower than body temperature for body parts such as knee joints. 

In conclusion, we pioneeringly employed trilayer graphene scrolls as FET channel material for a self-powered tribotronics and mechanosensation matrix. This novel structure of trilayer graphene scrolls transistor endowed the triboelectronic FET with extremely stretchable properties (up to 120% strain parallel to the direction of charge transport and the ability to retain 35% of its original current output) compared to intrinsic graphene and stacking graphene sheet; excellent temperature sensitivity (precision as low as 1); and high transparency. Meanwhile, for tribotronics GSFET, both the output performances powered by distance-dependent tribopotential were in good agreement with applied gate voltage, verifying that tribotronics could serve as an effective power source to drive a mechanosensation active matrix. The mechanosensation matrix possessed tactile sensing properties of high sensitivity (1.125 mm^−1^), rapid response time (~16 ms), and an endurability operation over thousands of cycles. Owing to all these merits, this graphene scrolls mechanosensation active matrix were spatially distance and temperature dual-mapping and achieved real-time monitoring. We can demonstrate that this device holds promising prospects for self-powered mechanosensation matrix for application of for advanced disease diagnosis, therapy, and health monitoring.

## Figures and Tables

**Figure 1 nanomaterials-13-00528-f001:**
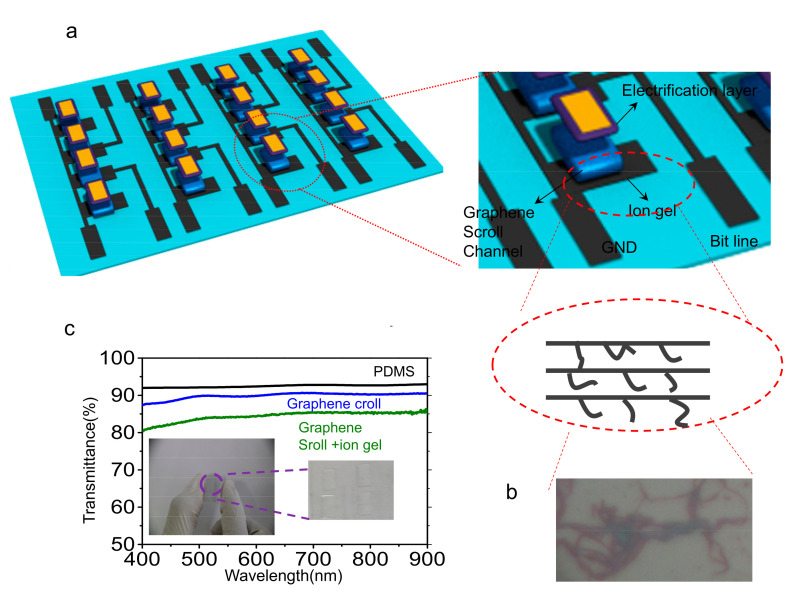
(**a**) Schematic illustration of the noncontact mechanosensation active matrix based on the tribotronic planar graphene transistors array. Inset is the zoomed-in schematic diagram of the single sensing unit. (**b**) Structure and microscopy image of the trilayer graphene scrolls. (**c**) UV-vis transmittance spectrum of PDMS, trilayer graphene scrolls on PDMS, and GSFET on PDMS, respectively. Inset is the photo image of transparent trilayer graphene scrolls.

**Figure 2 nanomaterials-13-00528-f002:**
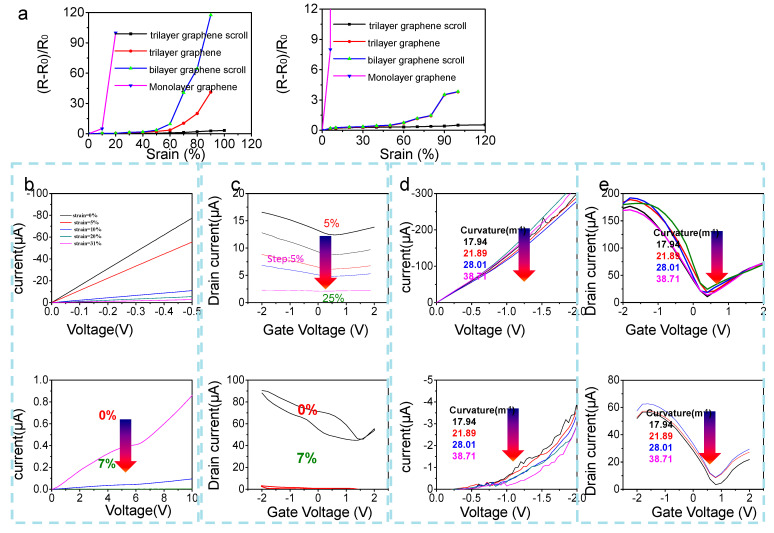
Electrical properties of GFET/GSFET under deformation. (**a**) Normalized relative resistance change of monolayer graphene, bilayer graphene scrolls, trilayer graphene, and trilayer graphene scrolls as a function of perpendicular (**left** panel) and parallel (**right** panel) stretch and strain to the direction of current flow. (**b**) *I**-V* curve of trilayer graphene scrolls (**top** panel) and monolayer graphene (**bottom** panel) at varied strain deformations. (**c**) Transfer curve of trilayer graphene scrolls (**top** panel) and monolayer graphene (**bottom** panel) at varied strain deformations (at given *V*_D_ of 0.1 V). (**d**) *I–V* curve of trilayer graphene scrolls (**top** panel) and monolayer graphene (**bottom** panel) at varied bending deformations. (**e**) Transfer curve of trilayer graphene scrolls (**top** panel) and monolayer graphene (**bottom** panel) at varied bending deformations.

**Figure 3 nanomaterials-13-00528-f003:**
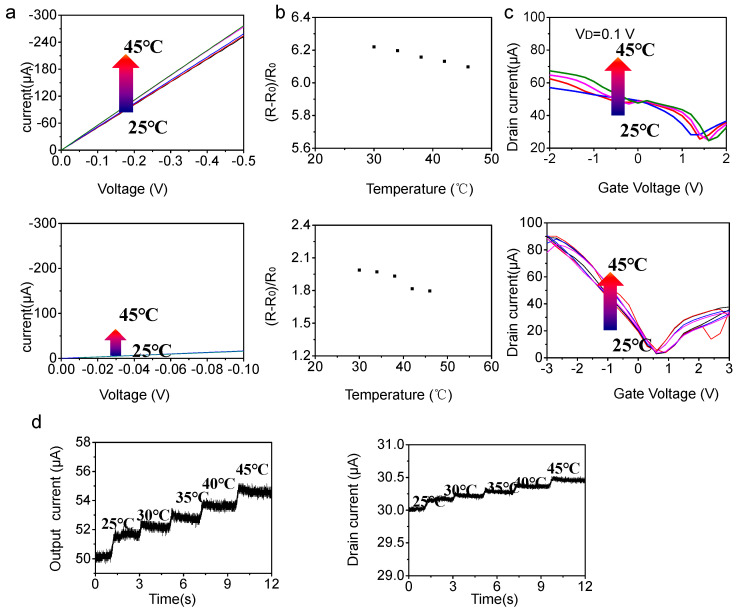
(**a**) The IV character of trilayer graphene scrolls (**top** panel) and monolayer graphene (**bottom** panel) (graphene films transferred on Ecoflex were patterned into 500 μm-wide and 2000 μm-long channels by photolithography). (**b**)The extracted relative resistance variation (*R* − *R*_0_)/*R*_0_ of trilayer graphene scrolls (**top** panel) and monolayer graphene (**bottom** panel). (**c**) Transfer characteristic of GSFET (**top** panel) and GFET (**bottom** panel) under varied temperature. (**d**) The real-time temperature monitoring of GFET (**right** panel) and GSFET (**left** panel).

**Figure 4 nanomaterials-13-00528-f004:**
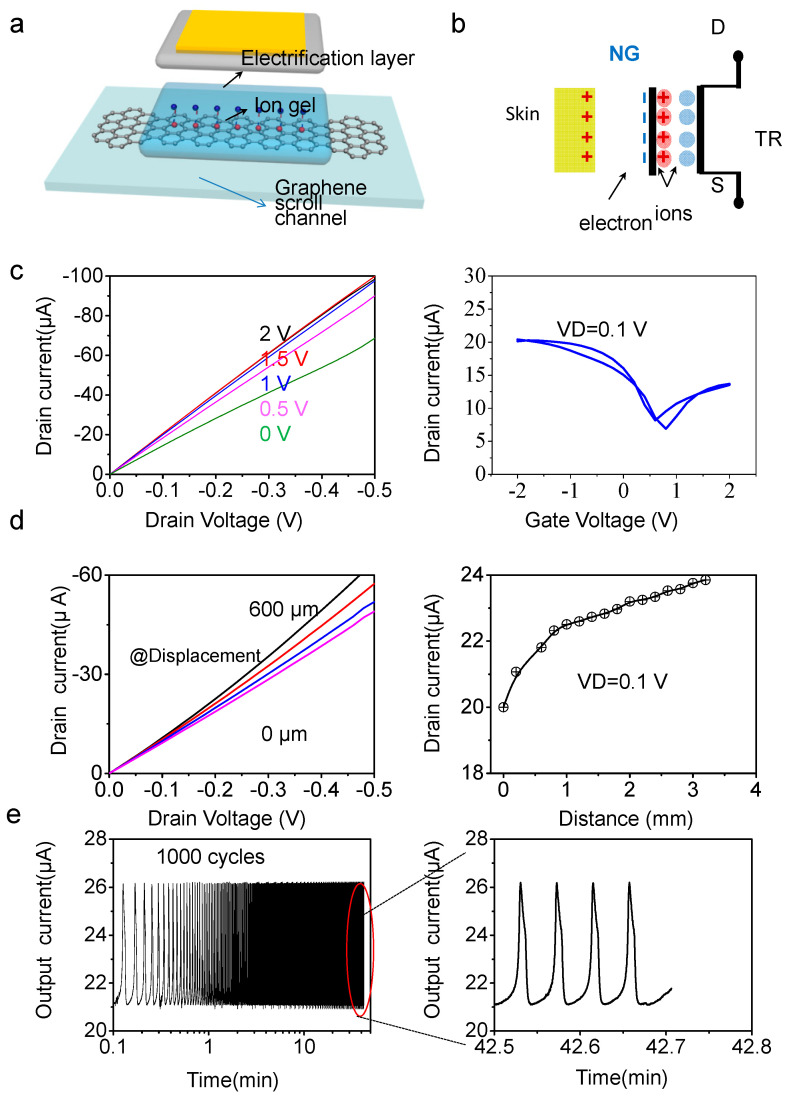
Triboelectric properties of the tribotronic GSFET. (**a**) Schematic illustration of the GSFET device. (**b**) Circuit diagram of tribotronic GSFET sensor contact with skin. (**c**) Output characteristic (left panel) and transfer characteristic (right panel) of GSFET by triboelectric driving. (**d**) Output characteristic (left panel) and transfer characteristic (right panel) of trilayer graphene scroll transistor by tribotronic potential (right panel). (**e**) Durability test over 3000 cycles of contact and separation (left panel) and retentivity during the last cycles (right panel). For E-skin application, ion gel dielectric was carried out friction with skin. Figure 4b shows the scheme of the working principle of ion gel-triboelectric gated GSFET. The tribo-potential, induced by friction between the ion gel and skin, causes the accumulation of negative charges on the surface of the ion gel, attracting compensating cations in the ion gel, while the anions in the ion gel migrate to the ion gel/graphene interface, acting as a negative gating potential for the graphene channel. Consequently, the electrons accumulate in graphene channel. The output characteristic of GFET under an applied gate (Figure 4c left panel) displays that the drain current (*I*_D_) exhibits a linear relationship with drain voltage *V*_DS_ and increases values as gate voltage *V*_G_ increases (from 68.7 μA to 98.5 μA with *V*_G_ increased from 0 V to 2 V at a *V*_D_ of 0.5 V). The transfer curve (Figure 4c right panel) displays that the *I*_D_ values increased with increased *V*_G_ in both positive and negative directions, indicating an ambipolar charge transport property of graphene. The GFET operated at a low gate voltage (<2 V) due to the extreme high capacitance of the ion gel gate dielectric (6–7 μF/cm^2^).

**Figure 5 nanomaterials-13-00528-f005:**
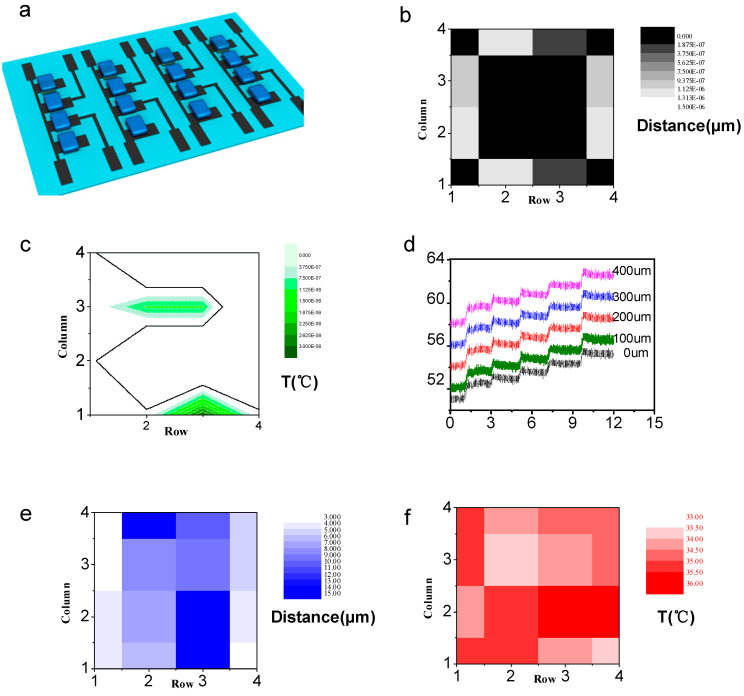
Characterization of GSFET noncontact mechanosensation active matrix. (**a**) Schematic illustration of the mechanosensation matrix for sensing the distance information and temperature. (**b**) 2D mapping of the distance sensing output currents for the matrix. (**c**) 2D mapping of the temperature sensing output currents for the matrix. (**d**) The real-time temperature monitoring and distance monitoring between the ion gel and skin. (**e**) 2D mapping of the distance sensing output currents for the matrix as the device is attached onto the human’s knee joint and conforms to it. (**f**) 2D mapping of the temperature-sensing output currents for the matrix as the device is attached onto the human’s knee joint and conforms to it.

## Data Availability

The data is available on reasonable request from the corresponding author.

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
