# Peer review of "Highly Stretchable Graphene Scrolls Transistors for Self-Powered Tribotronic Non-Mechanosensation Application"

_nanomaterials, 2023, doi:10.3390/nano13030528_

Round 1

Reviewer 1 Report

The manuscript entitled Highly stretchable Graphene scrolls Transistors for Self-Powered tribotronic non-Mechanosensation Application by Y. Meng demonstarted the stretchable tribotronic transistor using graphene materials. The manuscript is well written and supported with propoer experimental evidences, and the manuscript may be accepted with minor revision. 

1. The author should include few structural characterization experimental data other than UV-VIS to suport figure 1, and the fabricated graphene scrolls transistor.

2.  The cyclic stabililty should be  performed for a extended period of time till 2000 cycles

3. As the title shows its self-powered and there is no application on self-powered concept.

4. The authors should demonstarte a small application using the tribotronic transistor.

Author Response

We thank the referee for the positive comments. We have modified the manuscript substantially following the point-wise comments

Reviewer 2 Report

This manuscript reports on tribotronic mechanosensation active matrix based on tribotronic ion-gel graphene scrolls field-effect transistors (GSFET) which are transparent and flexible, enabling it operates with high sensitivity (0.08mm-1), rapid response time (~16ms) and a durability operation of thousands of cycles. The GSFET was used to sense touch stimuli and monitoring real-time temperature. Here are a few issues need to be addressed before its further consideration:

Major comments:

1.       This report studies mainly on graphene scroll for TENG that is different with previous report for single layer grapghene [Bao et al]. Figure 1b shows an optical microscope of trilayer graphenes but it doesn’t match with explanation of trilayer graphene scrolls. Can author explain where are graphene layers in this figure or provide new image for this issue?

2.       In figure 2b. Both left and right panels show the strain deformations were claimed similarly behavior when measured resistance in perpendicular and parallel directions, but it is clearly different in values and trend. In particular, trilayer graphene scrolls was broken at 100% left panel and 120% in right panel, and trilayer graphene and bilayer graphene scrolls show exactly same values in right panel.

3.       Author emphasized high performance that are high sensitivity (0.08mm-1) and rapid response time (~16ms) as highlights of this manuscript in abstract and discussion. However, author does not show detail of result and discussion for these points.

4.       IV curves and transfer curves of graphene and tri-layer graphene scrolls were performed with dependance of strain and bending deformations in figure 2. The hysteresis is one of graphene characterization that is relative to defect and doping, here by strain and bending deformation, the defect could be occurred and changed graphene properties. Therefore, the hysteresis probably influences on GSFET, here current could be different at different time of measurement with external electrical field. For temperature sensor that can affect mainly to be inaccurate. That points should be discuss and made clearly.

Mino comments:

5.       The body main text should use as same font and size.

6.       The axis range should be same (for example: Fig 2c and 2e). data and words in figure should be separated (Fig 3d)

7.       The manuscript has many errors, these should be corrected.

Author Response

We thank the referee for the positive comments. We have revised the manuscript substantially by including additional descriptions, modifying the grammatical mistake, and so on.
